# Effectiveness of Postnatal Maternal or Caregiver Interventions on Outcomes among Infants under Six Months with Growth Faltering: A Systematic Review

**DOI:** 10.3390/nu16060837

**Published:** 2024-03-14

**Authors:** Ritu Rana, Barkha Sirwani, Saranya Mohandas, Richard Kirubakaran, Shuby Puthussery, Natasha Lelijveld, Marko Kerac

**Affiliations:** 1Department of Public Health Programmes, Indian Institute of Public Health Gandhinagar, Gandhinagar 382042, Gujarat, India; barkhasirwani866@gmail.com (B.S.); saranyamohandas26leo@gmail.com (S.M.); 2Prof BV Moses Centre for Evidence Informed Health Care, Christian Medical College, Vellore 632004, Tamil Nadu, India; richrichigo@gmail.com; 3Maternal & Child Health Research Centre, Faculty of Health and Social Sciences, University of Bedfordshire, Luton LU1 3JU, UK; shuby.puthussery@beds.ac.uk; 4Emergency Nutrition Network, Kidlington OX5 2DN, UK; natasha@ennonline.net; 5Department of Population Health, London School of Hygiene and Tropical Medicine, London WC1E 7HT, UK; marko.kerac@lshtm.ac.uk

**Keywords:** infant, maternal, growth faltering, malnutrition, growth failure, wasting, underweight, failure to thrive, newborn, education intervention, nutrition

## Abstract

The care of infants at risk of poor growth and development is a global priority. To inform new WHO guidelines update on prevention and management of growth faltering among infants under six months, we examined the effectiveness of postnatal maternal or caregiver interventions on outcomes among infants between 0 and 6 months. We searched nine electronic databases from January 2000 to August 2021, included interventional studies, evaluated the quality of evidence for seven outcome domains (anthropometric recovery, child development, anthropometric outcomes, mortality, readmission, relapse, and non-response) and followed the GRADE approach for certainty of evidence. We identified thirteen studies with preterm and/or low birth weight infants assessing effects of breastfeeding counselling or education (*n* = 8), maternal nutrition supplementation (*n* = 2), mental health (*n* = 1), relaxation therapy (*n* = 1), and cash transfer (*n* = 1) interventions. The evidence from these studies had serious indirectness and high risk of bias. Evidence suggests breastfeeding counselling or education compared to standard care may increase infant weight at one month, weight at two months and length at one month; however, the evidence is very uncertain (very low quality). Maternal nutrition supplementation compared to standard care may not increase infant weight at 36 weeks postmenstrual age and may not reduce infant mortality by 36 weeks post-menstrual age (low quality). Evidence on the effectiveness of postnatal maternal or caregiver interventions on outcomes among infants under six months with growth faltering is limited and of ‘low’ to ‘very low’ quality. This emphasizes the urgent need for future research. The protocol was registered with PROSPERO (CRD42022309001).

## 1. Introduction

Early life growth faltering (also referred to as failure to thrive) and malnutrition are major global public health problems [1,2]. Affected infants include several distinct but often overlapping subpopulations: infants born at-risk (e.g., preterm, low birth weight (LBW), and/or intra-uterine growth failure); infants with subsequent growth faltering and associated anthropometric deficits (e.g., low weight-for-age, weight-for-length) [3,4]. Globally, each year, some 14.6% (20.5 million) of births are LBW and 10·6% (14.8 million) babies are born preterm [5,6]. These babies are at a high risk of mortality, and if they survive, are at risk of remaining or becoming malnourished during infancy [7].

Infants < 6 m with growth faltering are a particularly vulnerable group and are variously referred to as “infants at risk of poor growth and development” [8] or infants who are “small and nutritionally at-risk” [3]. These first six months of life are a period of rapid maturation and development with unique dietary needs: infants should ideally be breastfed during this period [9]. The mother or carer thus plays a critical role in fulfilling the nutritional requirements. If unmet, this can have serious implications for survival and later health. Short-term implications include a higher risk of morbidity and mortality, while long-term adverse effects include an elevated risk of non-communicable diseases and suboptimal cognitive development [10,11,12].

The care of infants < 6 m with early growth faltering is a global priority. The 2013 World Health Organization (WHO) guidelines for the Management of Severe Acute Malnutrition included, for the first time, a chapter on infants < 6 m but the recommendations in that guideline were based on limited and very low quality evidence [13]. Furthermore, most recommendations focused on infants’ needs, and did not describe details of wider factors that affect infants. Maternal or caregiver focused interventions were not formally examined. Since then, growing evidence is emerging which shows that numerous maternal factors are also associated with infant growth, nutrition and health outcomes [14].

To inform new guidelines and fill important care gaps to improve the treatment and prevention of malnutrition, in 2020, WHO began updating guidelines for the prevention and treatment of wasting. One of four focus areas is updating recommendations for infants < 6 m with growth faltering (others being the prevention of wasting; the management of moderate wasting; the management of severe wasting for children 6–59 months) [15]. Providing evidence to inform and shape these new WHO guidelines, this review aims to examine the effectiveness of postnatal maternal or caregiver interventions on outcomes among infants < 6 m with growth faltering.

## 2. Materials and Methods

### 2.1. Protocol and Ethics

This systematic review protocol was registered with the International Prospective Register of Systematic Reviews (PROSPERO) with registration number CRD42021276022. It was carried out following the Cochrane methodology and adhering to the Preferred Reporting Items for Systematic Reviews and Meta-Analyses (PRISMA) guidelines [16]. Ethical approval was not necessary for this study due to its nature.

### 2.2. Eligibility Criteria

Studies and setting: we included interventional studies—randomised controlled trials (RCTs) and non-randomised controlled trials (NRCTs), conducted in low-, middle-, and high-income countries.

Population: we included studies where the target population was mothers or caregivers (i.e., primary caregiver, father, or grandmother) of infants < 6 m with growth faltering. We defined growth faltering as

(i)wasting (weight-for-length z scores < −2 (WHO 2006)) or(ii)underweight (weight-for-age z scores < −2 (WHO 2006)) or(iii)low mid-upper arm circumference (MUAC) or(iv)small size at birth (preterm (<37 weeks of gestation) or(v)small for gestational age at birth (<10th centile) or(vi)low birth weight (<2500 g) or(vii)losing weight (study authors’ definition).

We excluded studies with specific groups such as babies born with certain congenital conditions.

Interventions and comparison: We included studies that assessed the effect of the following six intervention categories, (i) breastfeeding counselling or education, (ii) postnatal maternal nutrition supplementation, (iii) mental health, (iv) relaxation therapy, (v) cash transfers, and (vi) women empowerment. We excluded studies where the focus was not on postnatal maternal interventions, such as antenatal or prenatal interventions. We included studies where the comparison group did not include any of the aforementioned postnatal maternal-directed or caregiver interventions and/or compare interventions (or combinations) to each other. Comparison groups with standard care were included.

Outcomes: We included studies if they reported at least one of the following seven outcome domains among infants < 6 m with growth faltering: (i) anthropometric recovery (underweight or not, wasted or not), (ii) child development, (iii) anthropometric outcomes (change in anthropometric indices, weight gain, MUAC), (iv) mortality, (v) readmission, (vi) relapse, and (vii) non-response. We excluded studies with outcomes reported only for mothers or caregivers but not for infants < 6 m.

### 2.3. Information Source and Search Strategy

We conducted the search in nine electronic databases from January 2000 to August 2021: PubMed (MEDLINE), CINAHL Plus, Cochrane Library, EMBASE, Global Health, PsycINFO, Science Direct, Scopus, and Web of Science. Additionally, we also searched BIOSIS Previews, ISRCTN Registry, ICTRP WHO, and ClinicalTrials.gov. We used predefined search (title/abstract), MeSH terms, text words, and word variants for population and intervention terms to develop a comprehensive search strategy for each database. To identify randomised trials, we used different RCT filters for different databases [17]. The complete search strategy is presented in Appendix A.

### 2.4. Data Collection and Analysis

Selection process: We imported identified records into the EPPI-Reviewer Web (v 6.15.0.0) software [18]. Duplicate records were removed; first automatically and then manually. Two reviewers independently assessed records and reports at screening and eligibility stages, respectively, and any disagreements were resolved through discussion, or if required, consulting a third reviewer.

Data extraction and management: Two reviewers independently extracted data using an adapted electronic data extraction form from Cochrane [19]. The following data items were extracted for each trial: authors, year of publication, country, study characteristics, population characteristics, details of interventions, and outcome measures, including follow-up measurements at 9–12–18–24 months (if reported additionally). Any discrepancies in extracted data were resolved by discussion or by involving a third reviewer. Initially, the extracted data were recorded in an excel spreadsheet. Finally, we entered outcome data into Cochrane’s Review Manager (RevMan v 5.4.1, Cochrane Collaboration, London, UK) software and checked for accuracy [20].

### 2.5. Assessment of Risk of Bias

We evaluated the risk of bias at the outcome level in the included studies using the Revised Cochrane Risk of Bias tool for randomized trials (RoB 2) and the Risk Of Bias In Non-randomized Studies of Interventions (ROBINS-I) tool [21,22]. For RCTs, we assessed five key domains likely to influence results: the randomization process, deviations from intended interventions, missing outcome data, outcome measurement, and selection of reported results. Our assessments within each domain led to an overall judgment of the risk of bias for the outcome under consideration, categorized as ‘low risk’, ‘some concern’, or ‘high risk’.

For NRCTs, we assessed the risk of bias on seven domains. These domains are classified under three categories: pre-intervention—which involves bias due to confounding, and selection bias; at intervention—bias in classification of interventions; and post-intervention—these are similar to the last four bias domains used in RCTs. With the help of signalling questions, we judged the risk of bias within each domain. The judgements within each domain steered to an overall risk of bias for the outcome being assessed—‘low’, ‘moderate’, ‘serious’ or ‘critical’.

Two independent reviewers conducted bias assessments and a third reviewer double-checked for accuracy. When differences in the assessment of the risk of bias existed, a consensus was reached by discussion. Because the reviewers were well-acquainted with the studies, it was impossible to maintain blinding during the assessment process regarding the study author, journal of publication, or results.

### 2.6. Measures of Treatment Effect

We have presented continuous outcomes as mean difference (MD) with 95% confidence interval (CI) or as standardised mean difference (SMD) as appropriate (wherever the studies have reported outcomes using different scales). For dichotomous outcomes, we have presented the results as risk ratio (RR) or risk difference with 95% CI.

### 2.7. Assessment of Heterogeneity

We examined the diversity of outcomes in different studies using the *I*^2^ quantity, which measures the proportion of variation between studies attributed to differences rather than random chance. A 0% value suggests no diversity observed, while values of 50% or higher indicate significant diversity. We planned to undertake both a fixed-effect and a random-effects meta-analysis, to identify and present the best-fit model. We used fixed effect model if there were single studies where the effect estimates were very large and pulled the estimate to a larger effect.

### 2.8. Assessment of Reporting Biases

We have fewer than ten studies contributing to a single comparison. Therefore, the construction of a funnel plot was not carried out. We conducted a comprehensive search for identifying all the published and unpublished studies.

### 2.9. Data Synthesis

We performed statistical analyses using the RevMan V.5.4.1 software [20]. We analysed outcomes on an intention-to-treat basis. If data for similar outcomes from two or more separate studies were available, we combined the data in a meta-analysis and a typical MD or RR with associated 95% CI. Other data are presented as narrative results.

### 2.10. Subgroup and Sensitivity Analysis

We planned to conduct subgroup analysis as per protocol; however, we did not find enough studies as well as the data for any of the predetermined subgroup. To assess the robustness of results, we conducted a planned sensitivity analysis based on type of study designs (RCTs vs. NRCTs), where applicable.

### 2.11. Certainty of Evidence

We used the Grading of Recommendations Assessment, Development, and Evaluation (GRADE) approach to assess the quality of evidence and grade the strength of recommendations for seven outcome domains [23]—anthropometric recovery (critical), child development (critical), anthropometric outcomes (critical), mortality (important), readmission (important), relapse (important), and non-response (important). For each possible outcome, we evaluated the confidence level of the body of evidence across four areas: (1) risk of bias, (2) inconsistency, (3) indirectness, and (4) imprecision, assigning ratings of ‘high’, ‘moderate’, ‘low’, or ‘very low’. If there were ‘serious’, ‘very serious’, or ‘extremely serious’ issues in any of these domains, the quality of evidence was downgraded by one, two, or three levels, respectively. We created GRADE evidence profiles for comparisons using RCTs where applicable and included detailed explanations in the notes.

## 3. Results

### 3.1. Study Flow

Search results and the process of selection are presented in Figure 1. We identified 17,108 records. After de-duplication and screening of records, 682 reports were assessed for eligibility. Of these, 640 reports were excluded. Finally, 42 studies were eligible for inclusion in the analysis; of these, two studies presented the same anthropometric data from a single trial [24,25], and henceforth were considered as a single study. Further, we excluded studies with kangaroo mother care (*n* = 28) as these were already covered in another WHO guideline update—WHO recommendations for care of the preterm or low birth weight infant while this review was ongoing [26]. Thus, 13 studies contributed to final synthesis.

### 3.2. Characteristics of Included Studies

Table 1 presents the characteristics of included studies. The number of participants in individual studies ranged from 38 to 528. The target population were mainly mother–infant pairs (*n* = 12), and most studies were RCTs (*n* = 8). Of all included studies, 12 reported anthropometric outcomes, three readmission, and one reported child development and one mortality. None of the studies reported anthropometric recovery, relapse, and non-response. Based on the predefined criteria, we grouped the studies according to the intervention categories: we found eight studies on breastfeeding counselling or education, two on maternal nutrition supplementation, and one study each on mental health, relaxation therapy, and cash transfer. We did not find any study on women’s empowerment.

We found the following outcome measures under each outcome domain:Child development: child development scoreAnthropometric outcomes: weight, length, and head circumference at different time points; and change in weight and head circumference z scoresMortality outcome: mortality (different time points)Readmission outcome: readmission (different time points)

Of 13 studies reporting the aforementioned outcome measures, one did not report meaningful data [31]. For comparisons with two or more RCTs, we have presented the pooled estimates using a fixed-effect model, while other studies (less than two studies under a comparison, RCTs/NRCTs) are presented as narrative results.

### 3.3. Risk of Bias of Included Studies

We assessed the outcome level risk of bias for a total of 32 outcomes from RCTs and 10 outcomes from NRCTs. The results of risk of bias assessments are presented as Appendix A.

### 3.4. Summary of Included Studies

#### 3.4.1. Breastfeeding Counselling or Education

Eight studies were found on breastfeeding counselling or education interventions: four RCTs [24,27,28,33] and four NRCTs [29,30,31,32].

In the RCT from Iran [24], mothers with preterm infants (gestational age (GA) 34–37 weeks, and birth weight (BW) 2000–2500 g) received breastfeeding consultation sessions based on the BASNEF (Beliefs, Attitudes, Subjective Norms and Enabling Factors) model and counselling steps using GATHER (Greet clients, Ask clients about themselves, Tell clients about their choices, Help clients choose, Explain what to do, and Return for follow-up). Five hospital-based sessions (individual, face-to-face, 30 min) for five consecutive days were held by the researchers. The control group received conventional trainings by the staff.

In the RCT from Philippines [27], mothers with infants (GA 37–42 weeks, LBW) received homebased counselling visits (eight visits: at day 3–5, 7–10, 21, and at months 1.5, 2.5, 3.5, 4.5, 5.5) by peer counsellors trained in breastfeeding counselling. The training involved forty hours of interactive didactics, role-playing and practical training. The control group also received home visits (1st, 2nd, 3rd, and 6th month); however, the visits were for data collection only.

In another RCT from Iran [28], mothers with infants (GA < 37 weeks, BW < 2500 g) received a home visit training programme—the 1st day after discharge (20 min, difference between term and preterm baby, how to take care of preterm baby); the next day (20 min, advantages of breastfeeding); after 6 days (bath care); and the 1st, 2nd, 3rd, and 6th month (breastfeeding techniques). The intervention was provided by a paediatric nurse. The control group received visits for data collection on the 1st, 2nd, 3rd, and 6th month.

In an NRCT from South Korea [29], mothers with infants (GA 34–37 weeks) received late-preterm infant care education programme by researchers. Four sessions were held—within 1–2 days after childbirth (60 min, characteristics of late preterm infant, lecture/demonstration/practice), 3–4 days after childbirth (30 min, breastfeeding for late preterm infant, lecture/demonstration/practice), a day before the baby was discharged (40 min, post-discharge management of late preterm infant, lecture), and one per week for one month post discharge (10 min, emotional support, counselling). The control group received conventional education at admission, a day before discharge and on the discharge day.

In another NRCT from South Korea [31], mothers with infants (GA 34–37 weeks) received a breastfeeding support programme, which consisted of web-based breastfeeding education and practical breastfeeding support as per late-preterm infants feeding ability and mothers’ health issues at discharge and at four home visits (one/week for 4 weeks) by researchers. In the control group, mothers received counselling on nurturing skills for late-preterm infants. In both groups, mothers received home visits from the same researcher.

In an NRCT from Iran [30], mothers with infants (preterm) received five training sessions through telenursing by neonatal intensive care unit (NICU) nurse. The intervention group participated in a series of sessions covering various topics: breastfeeding technique and diet, changing the infant’s position and sleep patterns, skin and perineal care, medications and their complications, and dressing care, infection control procedures, and scheduling the next doctor’s visit. The control group, on the other hand, received standard care.

In another NRCT from Iran [32], mothers with infants (GA 34–37 weeks) received a supportive care program on breastfeeding behaviour. Four sessions of 60–90 min each (breast pumping, expressing breast milk, bathing of neonates, assessing and assisting breastfeeding practice) were provided. In the control group, mothers received the routine care and infants were followed up for two months.

In the RCT from Bangladesh [33], mothers with infants (BW < 2500 g) received nutrition education twice weekly for two months on initiation of breastfeeding within one hour, EBF and increasing their dietary intake.

#### 3.4.2. Maternal Nutrition Supplementation

Two studies reported maternal nutrition supplementation: one RCT [35] and one NRCT [34].

The RCT from Canada [35] evaluated the effect of Docosahexaenoic acid (DHA) supplementation to mothers of infants (GA 23–28 weeks) compared to placebo. Mothers in the intervention group received oral capsules providing 1.2 g/d of DHA within 72 h of delivery until their infants reached 36 weeks postmenstrual age (PMA), while mothers in other group received placebo capsules (a mix of 50% corn oils and 50% soy oils). The DHA and placebo capsules tasted and looked identical.

The NRCT from Brazil [34] evaluated the effect of zinc chelate supplementation to mothers of infants (GA ≤ 34 weeks) compared to no supplementation. Mothers in the intervention group received a recommended daily supplementation of 50 mg zinc chelate from the time point when the minimum dietary volume reached 100 mL/kg/d until the infant reached 40 weeks PMA. The control group did not receive any supplementation.

#### 3.4.3. Mental Health

A single RCT from Iran [36] included supportive counselling for mothers of infants with GA 28–33 weeks (6 sessions 45–60 min each from within the first 72 h of birth until the infant reached the 3rd week of life). The counselling sessions occurred within the first 72 h after birth, then again at the end of the first week, and continued twice a week until the third week. The focus was on actively listening to clients, understanding their mental concerns, showing empathy, providing information about preterm birth and its causes, discussing the condition of preterm infants, highlighting the negative effects of stress on mothers and babies, demonstrating baby massage techniques through videos, promoting clients’ abilities, teaching relaxation techniques, and offering emotional support and answers to their questions. This intervention was carried out by the researchers for three weeks before the mothers were discharged from the hospital. In contrast, mothers in the control group received standard care, which included breastfeeding training.

#### 3.4.4. Relaxation Therapy

A single RCT from Germany [37] assessed the effect of relaxation therapy; the parents or primary caregivers of infants (GA ≤ 30 weeks) in the intervention group received ‘family-centred music therapy—an interactive live-improvised music therapy’ twice a week from the 21st day of infant’s life until discharge from the hospital. After evaluating the parent–infant relationship, taking into account their musical heritage and cultural background, a personalized treatment plan was established. Infants and parents participated in music therapy sessions either with skin-to-skin contact or with the infant lying in an incubator. Meanwhile, mothers in the control group received standard care according to hospital policies.

#### 3.4.5. Cash Transfer

A single RCT based in the USA [38] assessed cash transfer (labelled financial transfer) intervention, where mothers of preterm infants received a transfer of $200 per week, up to a maximum of $600, to encourage their visits to the NICU for skin-to-skin contact, while the control group received standard care as per hospital policies.

### 3.5. Summary of Comparisons

#### 3.5.1. Effect of Breastfeeding Counselling or Education Versus Standard Care

Of the total eight studies, no study reported anthropometric recovery, child development, mortality, relapse, and non-response outcomes. Included studies contributed to anthropometric and readmission outcomes.

##### Anthropometric Outcomes

Five studies reported weight (g) at one month of age; three RCTs [24,28,33] and two NRCTs [29,32]. Among the RCTs, the interventions included breastfeeding consultation sessions based on the BASNEF model and counselling steps using GATHER, the home visit training programme, and breastfeeding nutrition education. The pooled estimate showed higher weight in infants of the breastfeeding counselling or education group at one month of age (MD 220.82; 95% CI 155.76 to 285.88, very low certainty, Table 2). The results from NRCTs also showed higher weight in infants of mothers who received the late-preterm infant care education programme (MD 387.50; 95% CI 165.09 to 609.91) [29] and in infants of mothers who received the supportive care program on breastfeeding behaviours (MD 0.12; 95% CI 0.08 to 0.16) [32].

Four studies reported weight (g) at two months, which included three RCTs [24,25,28,33] and one NRCT [32]. The pooled effect estimate of RCTs showed better weight for infants in the education group (MD 367.30; 95% CI 296.05 to 438.56, very low certainty, Table 2) [24,25,28,33]. A similar effect was also reported by the NRCT (MD 400; 95% CI 360 to 440, single study) [32]. The interventions included breastfeeding consultation sessions based on the BASNEF model and counselling steps using GATHER, the home visit training programme, breastfeeding nutrition education, and the supportive care program on breastfeeding behaviours.

Three studies, two RCTs [28,33] and one NRCT [29] reported length (cm) at one month of age. Among the RCTs, the intervention included the ‘home based training program’ and the hospital based ‘nutrition education on breastfeeding’. The pooled estimate showed higher length in infants of the breastfeeding counselling or education group (MD 0.66; 95% CI 0.24 to 1.07, very low certainty, Table 2). The NRCT with the ‘late preterm infant care education program’ also reported higher length in the education group (MD 1.20; 95% CI 0.09 to 2.31) [29].

##### Readmission

Two NRCTs reported infant readmission by two months of age. One study where the mothers received a late-preterm infant care education programme showed no difference (RR 0.35; 95% CI 0.01 to 8.12) [29], while the other study where mothers received training sessions through telenursing by an NICU nurse reported less risk of readmission (RR 0.39; 95% CI 0.25 to 0.61) [30]

#### 3.5.2. Effect of Maternal Nutritional Supplementation Versus Standard Care

No study reported anthropometric recovery, child development, readmission, relapse, and non-response outcomes. Included studies contributed to anthropometric and mortality outcomes.

##### Anthropometric Outcomes

Two studies (one RCT and one NRCT) observed the effect of maternal nutrition supplementation on anthropometric outcomes in preterm infants. The RCT [35] reported no difference in weight (g) by 36 weeks PMA between the infants of DHA supplementation group and the placebo group (MD −18.70; 95% CI −89.80 to 52.40, low certainty, Table 3). The NRCT [34] reported no difference in weight (MD 141.00; 95% CI −103.63 to 385.63), length (MD 0.00; 95% CI −1.53 to 1.53), or head circumference (MD 0.00; 95% CI −0.93 to 0.93) by 36 weeks PMA between the infants of the zinc chelate supplementation group and the control group.

##### Mortality Outcome

Only one RCT reported the effect of maternal nutrition supplementation intervention on mortality [35]. At 36 weeks PMA, there was no difference in the risk of infant mortality between mothers who received DHA supplementation and those who received the placebo (RR 0.61; 95% CI 0.33 to 1.13, low certainty, Table 3). Effect of mental health intervention versus standard care.

#### 3.5.3. Effect of Mental Health Intervention Versus Standard Care

##### Anthropometric Outcomes

A single RCT reported the effects of mental health intervention on anthropometric outcomes in a sample of 66 preterm infants [36]. At two months of age, there was no difference between the intervention and comparison group in weight (MD 0.30; 95% CI −355.14 to 355.74, very low certainty), length (MD −0.30; 95% CI −2.22 to 1.62, low certainty), or head circumference (MD 0.30; 95% CI −0.96 to 1.56, low certainty) (Table 4). The intervention group consisted of mothers who received supportive counselling.

#### 3.5.4. Effect of Relaxation Therapy Versus Standard Care

##### Anthropometric Outcomes

A single RCT [37] reported no difference between the two groups in weight (MD −42.06; 95% CI −242.25 to 158.13, very low certainty), length (MD 0.40; 95% CI −0.93 to 1.73, very low certainty) or head circumference (MD 0.07; 95% CI −0.63 to 0.77, very low certainty) at three months of age (Table 5). The intervention group consisted of parents or primary caregivers of infants (GA ≤ 30 weeks) who received ‘family-centred music therapy—an interactive live-improvised music therapy’.

#### 3.5.5. Effect of Cash Transfer Versus Standard Care

##### Child Development Outcomes

One RCT [38] reported the effect of cash transfer on the child development outcome (a composite of five survey items about infant development) at three months after discharge. Infants whose mothers received cash transfer (labelled financial transfer of $200 per week, up to a maximum of $600) had less development scores than infants in the comparison group (MD −1.05; 95% CI −1.62 to −0.48, very low certainty) (Table 6).

##### Anthropometric Outcomes

The same RCT [38] evaluated the effect of cash transfer for mothers of preterm infants on anthropometric outcomes. However, the study did not find any difference in the change in the weight z score (MD 0.58; 95% CI −0.23 to 1.39, very low certainty), and head circumference z score (MD −0.51; 95% CI −2.54 to 1.52, very low certainty) from birth to three months post discharge between the two groups (Table 6).

##### Readmission Outcome

The RCT also assessed the effect of cash transfer on readmission by three months of age [38]; there was no difference in the risk of being readmitted between the two groups (risk difference −0.22; 95% CI −0.51 to 0.07, very low certainty) (Table 6).

Results of outcomes other than critical and important for the guideline development process are presented as Appendix A.

## 4. Discussion

To inform new WHO guidelines for prevention and management of growth faltering among infants < 6 m, this review examined the effectiveness of postnatal maternal or caregiver interventions. These have an important potential role in improving infant survival, growth, development, and health and include breastfeeding counselling or education, nutrition supplementation, mental health, relaxation therapy, cash transfers, and women empowerment. Infants < 6 m outcomes included anthropometric recovery, child development, anthropometric outcomes, mortality, readmission, relapse, and non-response.

### 4.1. Summary of Evidence

Overall, few studies were identified—this itself is a notable finding given the global scale of the problem with millions of infants < 6 m worldwide affected. Despite a broad review question and inclusive search terms covering a wide range of different types of growth faltering and related conditions, only 13 studies were included in the current review, limiting our assessment to a narrative description of findings from the included studies with a few exceptions. We included eight studies on breastfeeding counselling or education, two on maternal nutrition supplementation, and one each on mental health, relaxation therapy, and cash transfer, while we did not find any studies on women empowerment. The included studies failed to examine the impact on anthropometric recovery, relapse, and non-response.

As well as being limited in extent, the overall body of evidence was also mostly ‘low’ to ‘very low’ certainty. We found that (i) breastfeeding counselling or education compared to standard care may increase infant weight at one month, weight at two months and length at one month; however, the evidence is very uncertain (very low quality). (ii) Maternal nutrition supplementation compared to standard care may not increase infant weight at 36 wks PMA and may not reduce infant mortality by 36 wks PMA (low quality). The evidence also suggests that (iii) maternal mental health interventions compared to standard care may not increase infant weight, length, and head circumference at two months (low to very low quality). (iv) The evidence is very uncertain about the effects of relaxation therapy compared to standard care on infant weight, length, and head circumference at three months (very low quality). (v) Furthermore, the evidence is very uncertain about the effects of cash transfers compared to standard care on child development scores at three months, change in weight and head circumference z score from birth to three months post-discharge, and readmission by three months (very low quality).

### 4.2. Overall Completeness and Applicability of Evidence

Most studies included mothers or caregivers of neonates, mainly preterm or LBW, leading to ‘serious’ concerns of indirectness. In nutrition and health programmes in settings where the new WHO guidelines on malnutrition are most needed, many other infants are also important, notably those who are underweight and/or wasted [39,40]. Some of these might also be ex preterm or LBW—but many will not [41]. It is possible but not certain that evidence arising from research conducted on neonates applies equally to those who develop growth faltering and anthropometric deficits later in infancy. The very same interventions may be more—or less—effective in these older infants. There is a notable lack of evidence directly focusing on the main future target group of underweight, wasted, and low MUAC infants. Growth trends are often unknown in LMIC settings so even growth faltering whereby at least two sequential measures are known is rarely known and is poorly interpreted even when documented [42].

We did not find data for key outcomes that are considered most important for the development of these particular guidelines. No studies reported anthropometric recovery (critical outcome), relapse (important outcome) and non-response (important outcome), which affects completeness of evidence. These are standard wasting programme outcome indicators. Furthermore, most outcomes from RCTs had ‘serious’ [28,33,37] to ‘very serious’ [38] risk of bias, and from NRCTs had ‘serious’ [29] to ‘critical’ [34] concerns of bias. These limitations affect the applicability and generalisability of the identified evidence. Lastly, most studies were from upper middle and high-income countries. Hence, it is uncertain if results will apply in the same way to settings where the WHO guidelines are most needed and will be most used: LMIC settings with important cultural, socioeconomic, and other contextual differences including a far larger burden of disease of early life malnutrition and growth faltering. Any generalizations must be made with great caution.

### 4.3. Review Findings in Context of Other Reviews

Although some previous reviews have investigated effectiveness of maternal breastfeeding counselling or education interventions on feeding outcomes, very few assessed early growth outcomes (<6 m). Where this was assessed, the evidence was limited [14,43,44,45]. Similarly, there is limited evidence to support the effect of maternal micronutrient supplementation for breastfeeding mothers, and cash transfers on infants < 6 m with growth faltering [46,47]. Other interventions have shown more promise. Maternal mental health interventions for example have potential, but the evidence from other reviews was also limited and weak [14]. Relaxation therapies also have many potential benefits [48]. They also have plausible mechanisms of action via increased milk supply, decreased breastmilk cortisol and thus improved infant growth as well as reduced maternal stress. However, more evidence is needed, especially evidence regarding effects on infants who are most at-risk, those with growth faltering and/or malnutrition. Overall, our review adds to previous ones highlighting many important research gaps.

### 4.4. Potential Biases in the Review Process

This review has several notable strengths, including having a protocol that was established before publication, a clearly defined research question, specific criteria for including and excluding studies, a thorough search strategy, the involvement of two reviewers in selecting studies and extracting data, an assessment of the risk of bias, and a rating of the certainty of evidence using the GRADE approach. In addition, we contacted study authors for clarity on the primary study data, where applicable. We acknowledge that we have one major deviation from the published protocol—exclusion of 28 studies on kangaroo mother care as these were already covered in other WHO guideline update (WHO recommendations for care of the preterm or low birth weight infant (2022)), while the review was ongoing [26]. This review is restricted to maternal or caregiver interventions during postnatal period with at least one infant outcome assessed <6 m. Thus, it is possible that interventions that started in the prenatal or antenatal period and continued until postal period could have been excluded, which might have provided valuable data.

### 4.5. Implications for Guidelines and Future Research

Based on the findings of our review, strong new WHO recommendations for maternal interventions for infants < 6 m growth faltering will be challenging to make because these need a more secure underpinning evidence base. Despite this, the need to support mothers is both plausible and has a strong moral/ethical case and hence there was much mention of the need to support mothers as well as infants < 6 m [15]. Both 2013 and 2023 WHO guidelines recommend provision of counselling and support, including mental health, to mothers or caregivers of infants < 6 m with severe malnutrition who are admitted for inpatient care and also for those who do not require inpatient care [13]. The guidelines recommended transfer of these infants to outpatient care if the mother or caregiver is linked to community-based follow-up and support. Given that 2023 guidelines are still based on ‘low’ to ‘very low’ quality evidence, with little change since 2013, the need for well-designed, well targeted future trials assessing postnatal maternal or caregiver interventions on infants < 6 m with growth faltering is more urgent than ever.

There are numerous issues that future trials should focus on. Firstly, they should include infants with growth faltering after and not just in the neonatal period (**population**). There is much current debate about optimal anthropometric criteria to identify these infants with options including low weight-for-age, low MUAC, low weight-for-length as well as more complex measures [40,49]. Until there are definitive answers, wide inclusion criteria would help.

Second, a wide range of possible **interventions** should be considered. It is unlikely that there will be any one ‘magic bullet’ and a package of interventions might thus be the most likely to succeed. Such packages have already been developed and are being tested [10,50]. Details of such packages will vary and indeed may have to vary in different contexts but are likely to include many interventions we explored in this review: maternal micro and macronutrient supplementation, mental health support, relaxation therapy, cash transfer, women empowerment: not just breastfeeding counselling or education which many practitioners and policymakers most strongly associate with this age group.

**Outcomes** will also have to be carefully considered in future work. Standard malnutrition and wasting program outcomes include anthropometric outcomes as well as death, default, and program transfer. However, it is vital to remember that anthropometry is not a perfect but a proxy measure of nutritional status: what matters far more is associated health (i.e., low risk of mortality and morbidity); child development (as assessed by well validated, locally appropriate tools [51]; and indeed clinically meaningful long-term post-program outcomes as well [52].

Finally, more future research needs to be done directly in LMIC and humanitarian settings where the burden of disease of infants < 6 m growth faltering/malnutrition is highest and the needs are greatest. This would minimize the many questions and dilemmas about generalizability that arise from the evidence based as identified in this review.

## 5. Conclusions

Evidence on the effectiveness of postnatal maternal or caregiver interventions on outcomes among infants < 6 m with growth faltering is limited and of ‘low’ to ‘very low’ quality. Due to lack of evidence on the population of interest, the studies included in this review had serious indirectness and their utility for future guidelines is limited. More focused, more directly applicable future research is urgently needed in this area.

## Figures and Tables

**Figure 1 nutrients-16-00837-f001:**
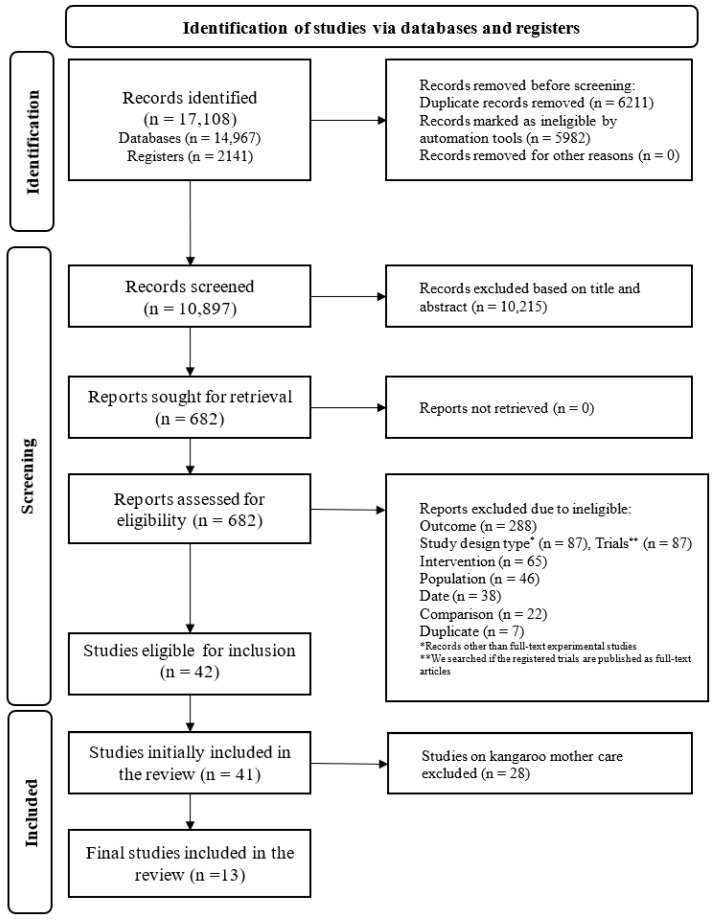
PRISMA flow diagram.

**Table 1 nutrients-16-00837-t001:** Characteristics of included studies.

Author (Year)	Country	Study Design (Sample Size)	Population	Eligible Outcomes Reported
Breastfeeding counselling or education				
Ahmadi (2016) [24]	Iran	RCT (124)	Mother–infant (GA 34–37 wks, BW 2000–2500 g)	Anthropometric
Agrasada (2005) [27]	Philippines	RCT (204)	Mother–infant (GA 37–42 wks, LBW)	Anthropometric
Edraki (2015) [28]	Iran	RCT (60)	Mother–infant (GA < 37 wks, BW < 2500 g)	Anthropometric
Eun Hye (2020) [29]	South Korea	NRCT (56)	Mother–infant (GA 34–37 wks)	Anthropometric, Readmission
Gholami (2021) [30]	Iran	NRCT (288)	Mother–infant (preterm)	Readmission
Gun Ja (2020) [31]	South Korea	NRCT (40)	Mother–infant (GA 34–37 wks)	Anthropometric
Moudi (2017) [32]	Iran	NRCT (84)	Mother–infant (GA 34–37 wks)	Anthropometric
Thakur (2012) [33]	Bangladesh	RCT (184)	Mother–infant (BW < 2500 g)	Anthropometric
Maternal nutrition supplementation				
de Figueiredo (2010) [34]	Brazil	NRCT (38)	Mother–infant (GA ≤ 34 wk)	Anthropometric
Marc (2020) [35]	Canada	RCT (528)	Mother–infant (GA 23–28 wks)	Anthropometric, Mortality
Mental health				
Seiiedi-Biarag (2021) [36]	Iran	RCT (66)	Mother–infant (GA 28–33 wks)	Anthropometric
Relaxation therapy				
Menke (2021) [37]	Germany	RCT (50)	Family-infant (GA ≤ 30 wks)	Anthropometric
Cash transfer				
Andrews (2020) [38]	USA	RCT (53)	Mother–infant (preterm)	Anthropometric, Child development, Readmission

BW: birthweight; g: gram; GA: gestational age; LBW: low birth weight; NRCT: non-randomized control trial; RCT: randomised control trial; USA: United States of America; wks: weeks.

**Table 2 nutrients-16-00837-t002:** GRADE evidence profile: breastfeeding counselling or education compared to standard care for mothers or caregivers of infants under six months with growth faltering.

Certainty Assessment						No. of Patients		Effect		Certainty	Importance
No. of Studies	Study Design	Risk of Bias	Inconsistency	Indirectness	Imprecision	Education	Standard Care	Relative (95% CI)	Absolute (95% CI)		
Anthropometric recovery (no data) [Critical]											
Child development (no data) [Critical]											
Weight (g) at 1 month (Anthropometric outcome)											
3 [24,28,33]	randomised trials	serious ^a^	serious ^b^	serious ^c^	not serious	184	184	-	MD 220.82 higher (155.76 higher to 285.88 higher)	⨁◯◯◯ Very low	Important
Weight (g) at 2 months (Anthropometric outcome)											
3 [24,28,33]	randomised trials	serious ^a^	serious ^d^	serious ^c^	not serious	184	184	-	MD 367.3 higher (296.05 higher to 438.56 higher)	⨁◯◯◯ Very low	Important
Length (cm) at 1 month (Anthropometric outcome)											
2 [28,33]	randomised trials	serious ^e^	serious ^f^	serious ^g^	not serious	122	122	-	MD 0.66 higher (0.24 higher to 1.07 higher)	⨁◯◯◯ Very low	Important
Mortality (no data) [Important]											
Readmission (no data) [Important]											
Relapse (no data) [Important]											
Non-response (no data) [Important]											

CI: confidence interval; MD: mean difference; ^a^ downgraded by one level for serious risk of bias. Two out of three studies have some concerns in risk of bias, including the study with highest weightage [33]; ^b^ downgraded by one level for serious inconsistency. The I sq statistics value is 68%, with one out of the three studies showing the opposite direction of effect; and ^c^ downgraded by one level for serious indirectness. Two [24,28] out of the three studies were from the same country (Iran). The target population among the studies were infants < 1 month of age (does not represent the whole 0–6 months age group) and the duration of education ranged from 2 months [33] to 6 months of infant’s age [28]; ^d^ downgraded by one level for serious inconsistency. The I sq value statistics is 91%; ^e^ downgraded by one level for serious risk of bias. Two studies contributing to the data have some concerns in risk of bias; ^f^ downgraded by one level for serious inconsistency. The I sq statistics value is 53%, with both the studies showing different directions of effect; ^g^ downgraded by one level for serious indirectness. The target population was not representative of the whole 0–6 months age group and the duration of intervention was until 2 months [33] or until 6 months of infant’s age [28]. Certainty rating: ‘(⨁⨁⨁⨁) high’, ‘(⨁⨁⨁◯) moderate’, ‘(⨁⨁◯◯) low’, or ‘(⨁◯◯◯) very low’.

**Table 3 nutrients-16-00837-t003:** GRADE evidence profile: maternal nutrition supplementation compared to standard care for mothers or caregivers of infants under six months with growth faltering.

Certainty Assessment						No. of Patients		Effect		Certainty	Importance
No. of Studies	Study Design	Risk of Bias	Inconsistency	Indirectness	Imprecision	Nutrition Supplementation	Standard Care	Relative (95% CI)	Absolute (95% CI)		
Anthropometric recovery (no data) [Critical]											
Child development (no data) [Critical]											
Weight (g) at 36 weeks PMA (Anthropometric outcomes)											
1 [35]	randomised trials	not serious	not serious	serious ^a^	serious ^b^	246	222	-	MD 18.7 lower (89.8 lower to 52.4 higher)	⨁⨁◯◯ Low	Important
Mortality by 36 weeks PMA											
1 [35]	randomised trials	not serious	not serious	serious ^a^	serious ^c^	16/268 (6.0%)	26/255 (10.2%)	RR 0.61 (0.33 to 1.13)	40 fewer per 1000 (from 68 fewer to 13 more)	⨁⨁◯◯ Low	Important
Readmission (no data) [Important]											
Relapse (no data) [Important]											
Non-response (no data) [Important]											

CI: confidence interval; MD: mean difference; RR: risk ratio; and ^a^ downgraded by one level for serious indirectness. The data comes from only one study [35] conducted in Canada with a sample size of 528 and inclusion criteria were mothers 16 years or older, intended to provide their breastmilk to their preterm infant, and were within 72 h of delivery at the time of randomisation (target population does not represent the whole 0–6 months age group); ^b^ downgraded by one level for serious imprecision. The 95% confidence interval is wide ranging from −89.8 to 52.4; ^c^ downgraded by one level for serious imprecision. The 95% confidence interval is wide ranging from 0.33 to 1.13. Certainty rating: ‘(⨁⨁⨁⨁) high’, ‘(⨁⨁⨁◯) moderate’, ‘(⨁⨁◯◯) low’, or ‘(⨁◯◯◯) very low’.

**Table 4 nutrients-16-00837-t004:** GRADE evidence profile: maternal mental health interventions compared to standard care for mothers or caregivers of infants under six months with growth faltering.

Certainty Assessment						No. of Patients		Effect		Certainty	Importance
No. of Studies	Study Design	Risk of Bias	Inconsistency	Indirectness	Imprecision	Mental Health Interventions	Standard Care	Relative (95% CI)	Absolute (95% CI)		
Anthropometric recovery (no data) [Critical]											
Child development (no data) [Critical]											
Weight at 2 months (Anthropometric outcomes)											
1 [36]	randomised trials	not serious	not serious	serious ^a^	very serious ^b^	32	30	-	MD 0.3 higher (355.14 lower to 355.74 higher)	⨁◯◯◯ Very low	Important
Length at 2 months (Anthropometric outcomes)											
1 [36]	randomised trials	not serious	not serious	serious ^a^	serious ^c^	32	30	-	MD 0.3 lower (2.22 lower to 1.62 higher)	⨁⨁◯◯ Low	Important
Head circumference at 2 months (Anthropometric outcomes)											
1 [36]	randomised trials	not serious	not serious	serious ^a^	serious ^d^	32	30	-	MD 0.3 higher (0.96 lower to 1.56 higher)	⨁⨁◯◯ Low	Important
Mortality (no data) [Important]											
Readmission (no data) [Important]											
Relapse (no data) [Important]											
Non-response (no data) [Important]											

CI: confidence interval; MD: mean difference; ^a^ downgraded by one level for serious indirectness. Only one study [36] conducted in Iran contributed to the data with a sample size of 66 infants born between 28 and 33 weeks and being 6 days hospitalized in NICU, enrolled within 72 h of birth (target population does not represent the whole 0–6 months age group); ^b^ downgraded by two levels for a very serious imprecision. The 95% confidence interval is wide ranging from −355.14 to 355.74; ^c^ downgraded by one level for serious imprecision. The 95% confidence interval is wide ranging from −2.22 to 1.62; ^d^ downgraded by one level for serious imprecision. The 95% confidence interval is wide ranging from −0.96 to 1.56. Certainty rating: ‘(⨁⨁⨁⨁) high’, ‘(⨁⨁⨁◯) moderate’, ‘(⨁⨁◯◯) low’, or ‘(⨁◯◯◯) very low’.

**Table 5 nutrients-16-00837-t005:** GRADE evidence profile: relaxation therapy compared to standard care for mothers or caregivers of infants under six months with growth faltering.

Certainty Assessment						No. of Patients		Effect		Certainty	Importance
No. of Studies	Study Design	Risk of Bias	Inconsistency	Indirectness	Imprecision	Relaxation Therapy	Standard Care	Relative (95% CI)	Absolute (95% CI)		
Anthropometric recovery (no data) [Critical]											
Child development (no data) [Critical]											
Weight at 3 months (Anthropometric outcomes)											
1 [37]	randomised trials	serious ^a^	not serious	serious ^b^	very serious ^c^	24	26	-	MD 42.06 lower (242.25 lower to 158.13 higher)	⨁◯◯◯ Very low	Important
Length at 3 months (Anthropometric outcomes)											
1 [37]	randomised trials	serious ^a^	not serious	serious ^b^	serious ^d^	24	26	-	MD 0.4 higher (0.93 lower to 1.73 higher)	⨁◯◯◯ Very low	Important
Head circumference at 3 months (Anthropometric outcomes)											
1 [37]	randomised trials	serious ^a^	not serious	serious ^b^	Serious ^e^	24	26	-	MD 0.07 higher (0.63 lower to 0.77 higher)	⨁◯◯◯ Very low	Important
Mortality (no data) [Important]											
Readmission (no data) [Important]											
Relapse (no data) [Important]											
Non-response (no data) [Important]											

CI: confidence interval; MD: mean difference; ^a^ downgraded by one level for serious risk of bias. The only study [37] contributing to the data has some concerns in the risk of bias; ^b^ downgraded by one level for serious indirectness. A single study [37] contributing to the data is conducted in a high-income country (Germany) and has a very small sample size. The target population is <1 month of age (does not represent the whole 0–6-month age group); ^c^ downgraded by two levels for a very serious imprecision. The 95% confidence interval is wide ranging from −242.25 to 158.13; ^d^ downgraded by one level for serious imprecision. The 95% confidence interval is wide ranging from −0.93 to 1.73; ^e^ downgraded by one level for serious imprecision. The 95% confidence interval is wide ranging from −0.63 to 0.77. Certainty rating: ‘(⨁⨁⨁⨁) high’, ‘(⨁⨁⨁◯) moderate’, ‘(⨁⨁◯◯) low’, or ‘(⨁◯◯◯) very low’.

**Table 6 nutrients-16-00837-t006:** GRADE evidence profile: cash transfer compared to standard care for mothers or caregivers of infants under six months with growth faltering.

Certainty Assessment						No. of Patients		Effect		Certainty	Importance
No. of Studies	Study Design	Risk of Bias	Inconsistency	Indirectness	Imprecision	Cash Transfer	Standard Care	Relative (95% CI)	Absolute (95% CI)		
Anthropometric recovery (no data) [Critical]											
Child development score at 3 months											
1 [38]	randomised trials	very serious ^a^	not serious	serious ^b^	not serious	0	0	-	MD 1.05 lower (1.62 lower to 0.48 lower)	⨁◯◯◯ Very low	Critical
Change in weight z-score from birth to 3 months post-discharge (Anthropometric outcomes)											
1 [38]	randomised trials	very serious ^a^	not serious	serious ^b^	serious ^c^	0	0	-	MD 0.58 higher (0.23 lower to 1.39 higher)	⨁◯◯◯ Very low	Important
Change in head circumference z- score from birth to 3 months post-discharge (Anthropometric outcomes)											
1 [38]	randomised trials	very serious ^a^	not serious	serious ^b^	serious ^d^	0	0	-	MD 0.51 lower (2.54 lower to 1.52 higher)	⨁◯◯◯ Very low	Important
Mortality (no data)											
Readmission by 3 months											
1 [38]	randomised trials	very serious ^a^	not serious	serious ^b^	very serious ^e^	0	0	not estimable	220 more per 1000 (from 70 fewer to 510 more)	⨁◯◯◯ Very low	Important
Relapse (no data) [Important]											
Non-response (no data) [Important]											

CI: confidence interval; MD: mean difference; ^a^ downgraded by two levels for a very serious risk of bias. The study [38] contributing the data has a high risk of bias; ^b^ downgraded by one level for serious indirectness. The only study [38] contributing the data is conducted in an HIC (United States of America) and has a small sample size of 53 infants (target population is <1 month of age, not representative of the whole 0–6-months age group); ^c^ downgraded by one level for serious imprecision. The 95% confidence interval is wide ranging from −0.23 to 1.39; ^d^ downgraded by one level for serious imprecision. The 95% confidence interval is wide ranging from −2.54 to 1.52; and ^e^ downgraded by two levels for a very serious imprecision. The 95% confidence interval is wide ranging from −0.51 to 0.07.

## Data Availability

Not applicable.

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
