# Peer review of "Effectiveness of Postnatal Maternal or Caregiver Interventions on Outcomes among Infants under Six Months with Growth Faltering: A Systematic Review"

_nutrients, 2024, doi:10.3390/nu16060837_

Round 1
Reviewer 1 Report
Comments and Suggestions for Authors
The authors composed a systematic review on the effectiveness of postnatal maternal or caregiver interventions on outcomes among infants under six months with growth faltering.
First, it should be acknowledged that the authors followed the rules of composing a systematic review very precisely, so the present manuscript is a nice example of a professionally written systematic review.
Second, it should be also admitted that the authors didn’t have much luck at identifying studies corresponding to the inclusion criteria into this systemic review. They were able to include 8 studies (3 RCTs and 5NRCTs) investigating the effect of breastfeeding counselling or education, which number allows some review conclusions to draw. (Here I should mention that in Table 1 two different references belong to the first study [24 and 25]. In my view, if a research group decides to publish the same data in two papers, they should not by honoured by mentioning both papers as reference.)
In contrast to breastfeeding counselling or education, only one, one and one studies were identified on mental health, relaxation therapy and cash transfer. I am pretty sure that tabulation of data published in one study should not fill pages in Nutrients, therefore I recommend Tables 4, 5 and 6 to be delated form the main text (and probably be transferred to the supplementary section of the paper). The same consideration may hold true also for Table 3, in that only one of the two studies identified on maternal nutrition supplementation dealt with.
Bearing in mind the limited new information gained by this systematic review approach, the Discussion seems highly overwritten (2.5 printed page in its present form). I recommend to cut the discussion by about half.
Finally, a minor but sensitive issue. In line 82 (and probably also elsewhere) the authors write “we included experimental studies”. In my understanding, the wording “experimental” should never be used in connection with studies on human being. Please, use “interventional”, or some other similar wording, instead of “experimental”.
In summary, the authors should consider the recommendation suggested above and modify their manuscript accordingly.
Comments on the Quality of English Language
Only minor editing of English language is required.
Author Response
Please see the attachment. (all three reviews together for ease of interpretation across all)

Reviewer 2 Report
Comments and Suggestions for Authors
Dear Authors,
Please find attached the review.
Kindest regards

Author Response

(The authors gave the same response as above.)

Reviewer 3 Report
Comments and Suggestions for Authors
This is an excellent review. It is easy to follow what the authors did and how they assessed the published studies on this important topic. The writing is impeccable and I found only one minor quibble (line 150 "while assessment) they may wish to change.
Author Response

(The authors gave the same response as above.)

Round 2
Reviewer 1 Report
Comments and Suggestions for Authors
I still feel that the paper is based on limited data and, consequently, pretty much overwritten in its present form. The considerations about the importance of grading aspects may be true in special journals devoted to EBM, but they are definitely less interesting for readers of a nutritional journal. I, for one, do not think that citing one single study justifies the construction of a table.
I understand that two other reviewers did not criticise the length of the discussion, but I still feel it to be unnecessary lengthy.